# Nicotinic acetylcholine receptor modulator insecticides act on diverse receptor subtypes with distinct subunit compositions

Wanjun Lu[1]☯, Zhihan Liu[1]☯, Xinyu Fan[1]☯, Xinzhong Zhang[2]☯, Xiaomu Qiao[1], Jia Huang[1]*

1 Ministry of Agriculture Key Laboratory of Molecular Biology of Crop Pathogens and Insects, Institute of Insect Sciences, Zhejiang University, Hangzhou, China, 2 Tea Research Institute, Chinese Academy of Agricultural Sciences, Hangzhou, China

☯ These authors contributed equally to this work.
* huangj@zju.edu.cn

**Data Availability Statement:** All relevant data are within the manuscript and its Supporting information files.

## Abstract

Insect nicotinic acetylcholine receptors (nAChRs) are pentameric ligand-gated ion channels mainly expressed in the central nervous system of insects. They are the directed targets of many insecticides, including neonicotinoids, which are the most widely used insecticides in the world. However, the development of resistance in pests and the negative impacts on bee pollinators affect the application of insecticides and have created a demand for alternatives. Thus, it is very important to understand the mode of action of these insecticides, which is not fully understood at the molecular level. In this study, we systematically examined the susceptibility of ten *Drosophila melanogaster* nAChR subunit mutants to eleven insecticides acting on nAChRs. Our results showed that there are several subtypes of nAChRs with distinct subunit compositions that are responsible for the toxicity of different insecticides. At least three of them are the major molecular targets of seven structurally similar neonicotinoids in vivo. Moreover, spinosyns may act exclusively on the α6 homomeric pentamers but not any other nAChRs. Behavioral assays using thermogenetic tools further confirmed the bioassay results and supported the idea that receptor activation rather than inhibition leads to the insecticidal effects of neonicotinoids. The present findings reveal native nAChR subunit interactions with various insecticides and have important implications for the management of resistance and the development of novel insecticides targeting these important ion channels.

## Author summary

Neonicotinoids and spinosyns account for approximately 24% and 3% of the world market value of insecticides, respectively. However, the negative effects of neonicotinoids on pollinators have led to the development of novel insecticides, such as sulfoxaflor, flupyradifurone and triflumezopyrim. Although all act via insect nicotinic acetylcholine receptors, their modes of action are not fully understood. Our work shows that these

**Funding:** J.H. was supported by the National Natural Science Foundation of China (32072496, https://www.nsfc.gov.cn) and Zhejiang Provincial Fund for Distinguished Young Scholars (LR19C140002, http://zjnsf.kjt.zj.gov.cn/portal). X. M. was supported by the National Natural Science Foundation of China (31802019, https://www.nsfc. gov.cn). The funders had no role in study design, data collection and analysis, decision to publish, or preparation of the manuscript.

**Competing interests:** The authors have declared that no competing interests exist.

insecticides act on diverse receptor subtypes with distinct subunit compositions. This finding could lead to the development of more selective insecticides to control pests with minimal effects on beneficial insects.

## Introduction

Chemical insecticides have been widely used to control pests in the agriculture, horticulture, and forestry industries as well as homes and cities. They have also played a vital role in preventing the spread of human and animal vector-borne diseases. However, insecticide resistance is a serious worldwide problem for invertebrate pest control, and more than 600 different insect and mite species have become resistant to at least one insecticide. In addition, at least one case of resistance to more than 335 insecticides/acaricides has been documented [1]. Therefore, there is great demand for effective insecticide resistance management (IRM) and the development of new pest control compounds. To address both issues, we need to determine the mode of action of insecticides, which is the molecular-level processes underlying the effects of insecticides [2].

A complete understanding of the mode of action of an insecticide requires knowledge of how it affects a specific target site within an organism. Although most insecticides have multiple biological effects, toxicity is usually attributed to a single major effect. For some insecticides, however, the exact molecular targets remain elusive. To ascribe whether a candidate protein is indeed the target for an insecticidal effect in vivo, it is not sufficient to demonstrate an in vitro biochemical interaction between an insecticide and a protein. Genetic evidence demonstrating an effect due to mutation of the candidate target must be obtained before a given protein can be identified as an insecticide target.

Neonicotinoids (acetamiprid, clothianidin, dinotefuran, imidacloprid, nitenpyram, thiacloprid, and thiamethoxam) are remarkably effective at controlling agricultural pests, ectoparasites and arthropod vectors [3]. They are taken up by the roots or leaves and translocated to all parts of the plant due to high systemic activity, making them effectively toxic to a wide range of sap-feeding and foliar-feeding insects. Thus, neonicotinoids account for 24% of the global insecticide market, which is the largest market share of all chemical classes [1]. They act selectively on insect nicotinic acetylcholine receptors (nAChRs) as agonists compared with the mammalian-selective nicotine. Spinosyns are a naturally derived, unique family of macrocyclic lactones that act on insect nAChR in an allosteric fashion. In addition, sulfoximine sulfoxaflor, butenolide flupyradifurone and mesoionic triflumezopyrim are three newly developed insecticides that are also nAChR competitive modulators [4]. It is expected that the market of all of the above nAChRs targeting insecticides that show excellent insect-to-mammalian selectivity will continue to grow. However, the molecular targets of neonicotinoids and other nAChR modulators remain unclear, mainly because the structure and assembly of native nAChRs in insects have not been clarified [5].

Cation-selective nAChRs are members of the Cys-loop ligand-gated ion channel superfamily responsible for rapid excitatory neurotransmission. The functional nAChRs are homo- or heteromeric pentamers of structurally related subunits arranged around a central ion-conducting pore [6]. Each subunit has an extracellular N-terminal domain that contains six distinct regions (loops A–F) involved in ligand binding, four C-terminal transmembrane segments (TM1–TM4) and an intracellular loop between TM3 and TM4. nAChRs are divided into α-subunits possessing two adjacent cystine residues in loop C, while those subunits without this motif are termed nonα subunits. In vertebrates, 17 nAChR subunits have been identified, and

they can coassemble to generate a diverse family of nAChR subtypes with different pharmacological properties and physiological functions. Insects have fewer nAChR subunits (10–12 subunits) according to the available genome data. Although coimmunoprecipitation studies have indicated potential associations of several subunits, the exact subunit composition of native insect nAChRs remains unknown [5]. Unlike the vertebrate counterparts, heterologous expression of genuine arthropod α and β subunits was not successful until two groups recently found that three ancillary proteins are essential for robust expression of arthropod nAChR heteromers [7, 8]. Thus, for a long time, researchers have used hybrid receptors with insect α subunits and mammalian/avian β subunits to study the interaction of insecticides and receptors. However, such alternatives may not faithfully reflect all features of native nAChRs [9].

In this study, we systematically examined the effects of eleven nAChR-targeting insecticides against ten (seven α and three β) *Drosophila melanogaster* subunit mutants. We found that there are multiple subtypes of receptors with distinct subunit compositions that are responsible for the toxicity of different insecticides. Artificial activation/inhibition of subunit-expressing neurons also mimicked insecticide poisoning symptoms in pests. Elucidating the molecular targets of these economically important agrochemicals and the assembly of native nAChRs will be very helpful for resistance management and ecotoxicological evaluations of beneficial insects, such as predators and pollinators.

## Results

### Generation of the nAChRβ1$^{R81T}$ mutant

We obtained all 10 nAChR knockout mutants from Yi Rao's laboratory and found that the KO of α4 and β1 was homozygous lethal. Previous studies [10, 11] and our results (Tables A-K in S1 Text) both showed that target-site resistance in nAChRs is mostly recessive to semirecessive, with the wild-type allele being dominant. Therefore, we used point mutation alleles of α4 and β1 in all experiments instead of heterozygous KO flies. The α4$^{T227M}$ mutant (*redeye*, *rye*) is a dominant-negative allele that causes a reduced sleep phenotype in flies [12]. An R81T mutation of nAChR β1 was found in neonicotinoid-resistant peach aphids and later in cotton aphids [13, 14]; therefore, we introduced a homologous mutation into the β1 locus of *Drosophila melanogaster* with CRISPR–Cas9–mediated homology-directed repair (HDR). The design of the gRNA target site and HDR template was reported, and the screening of successful R81T knock-in was performed under imidacloprid selection pressure and confirmed by direct DNA sequencing (Fig 1 and Fig A in S1 Text).

### nAChR mutants showed distinct resistance to multiple insecticides

We tested the effects of 10 nAChR mutants and some heterozygous mutants against 11 insecticides (Fig 2 and Tables A-K in S1 Text). The α1 mutant showed moderate levels of resistance to imidacloprid, thiacloprid, acetamiprid and triflumezopyrim, and the LC$_{50}$ resistance ratio (RR) was approximately 13.5–88.0. Its heterozygous mutant also showed low levels of resistance to these insecticides. In addition, it showed low but statistically significant increases in RR (2.7–3.7) to thiamethoxam, clothianidine, dinotefuran and nitenpyram. The α2 mutant also showed similar levels of resistance (17.2–48.5 in terms of RR) to imidacloprid, thiacloprid and triflumezopyrim. The α3 mutant showed small RR increases (2.7–5.5) to thiamethoxam, clothianidine, dinotefuran, nitenpyram, sulfoxaflor and flupyradifurone. The α4, α5, α6, α7 and β3 mutants are sensitive to almost all the tested insecticides. The obvious exception is the α6 homozygous mutant, which is resistant to spinetoram and has a RR of 42.8, although the heterozygous mutant is close to the wild type (RR 1.2). The β1 mutant exhibited medium to high resistance to all insecticides (23.9–398.3 in terms of RR) except spinetoram, and its

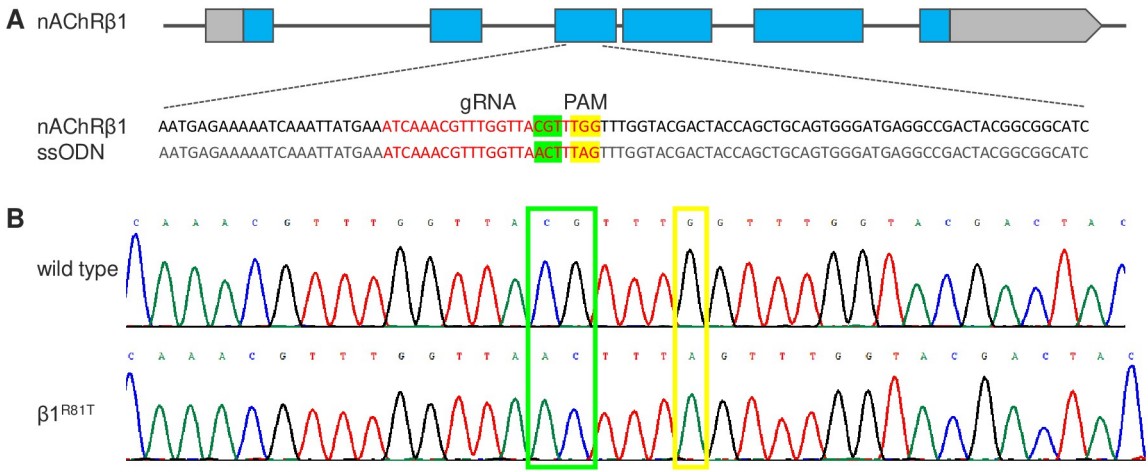

**Fig 1. Generation of the nAChRβ1^R81T mutant by CRISPR/Cas9 genome editing.** (A) Schematic of the nAChRβ1 locus and the sequence of the donor construct. The boxes represent exons, and the coding regions are shown in blue. The gRNA sequence is indicated in red, and the codon for amino acid substitution (CGT to ACT) is highlighted in green. One synonymous mutation (G to A) is also introduced in the PAM region (in yellow) to prevent recleavage from Cas9 after successful integration. (B) Sequence comparison between wild-type flies and flies with point mutations. The nucleotides replaced are highlighted in green and yellow boxes.

heterozygous mutant showed small RR increases for most insecticides. The resistance profile of the β2 mutant was similar to that of the α1 mutant, with a 13.0- to 84.3-fold increased RR to imidacloprid, thiacloprid, acetamiprid and triflumezopyrim.

Both the α1 and β1 mutants showed variable resistance to multiple insecticides; thus, we generated a α1/β1 double mutant with recombination. However, the eggs laid by this combined mutant could not hatch and thus were not used in further experiments. A recent paper also generated a β1 R81T *Drosophila* and found that it has serious defects in reproduction and locomotion [15]; however, the β1 mutant we made here did not show any significant fitness cost (Fig B in S1 Text). The sequences of α5, α6 and α7 are very close and show high similarity to the vertebrate nAChR α7 subunit; however, only the α6 mutant showed resistance to spinetoram. We further generated a α5/α7 double mutant that was still sensitive to spinetoram (Table K in S1 Text), indicating that the α6 homomeric channel could be the sole target for spinosyns.

## Hyperactivating/Silencing *nAChR*-expressing neurons mimics insecticide poisoning symptoms

Insects present similar reactions upon exposure to neonicotinoids, sulfoxaflor, flupyradifurone and spinosyns. Early-onset behaviors include hyperactivity, convulsion, uncoordinated movements, leg extension and tremors. At higher doses, these excitatory symptoms can induce severe tremors and complete paralysis, which ultimately leads to death [16–18]. We then wondered whether artificial activation of *nAChR*-expressing neurons would induce insecticide-like poisoning symptoms. Thus, we used the thermosensitive cation channel *Drosophila* TRPA1 to acutely hyperstimulate these neurons with all available *nAChR* KI-Gal4 strains [19]. We found that expressing *trpA1* in *nAChRα1^{2A-GAL4}*, *nAChRα2^{2A-GAL4}*, *nAChRα3^{2A-GAL4}*, *nAChRα6^{2A-GAL4}* and *nAChRβ2^{2A-GAL4}* neurons strongly induced hyperactivity behavior at 32°C and eventually led to paralysis (Fig 3A and S1 Video), which is similar to the abovementioned symptoms. However, activation of *nAChRβ3^{2A-GAL4}* neurons did not show any behavioral defects. These results parallel the above bioassay data showing that the deletion of α1, α2, α3, α6 and β2 caused medium to

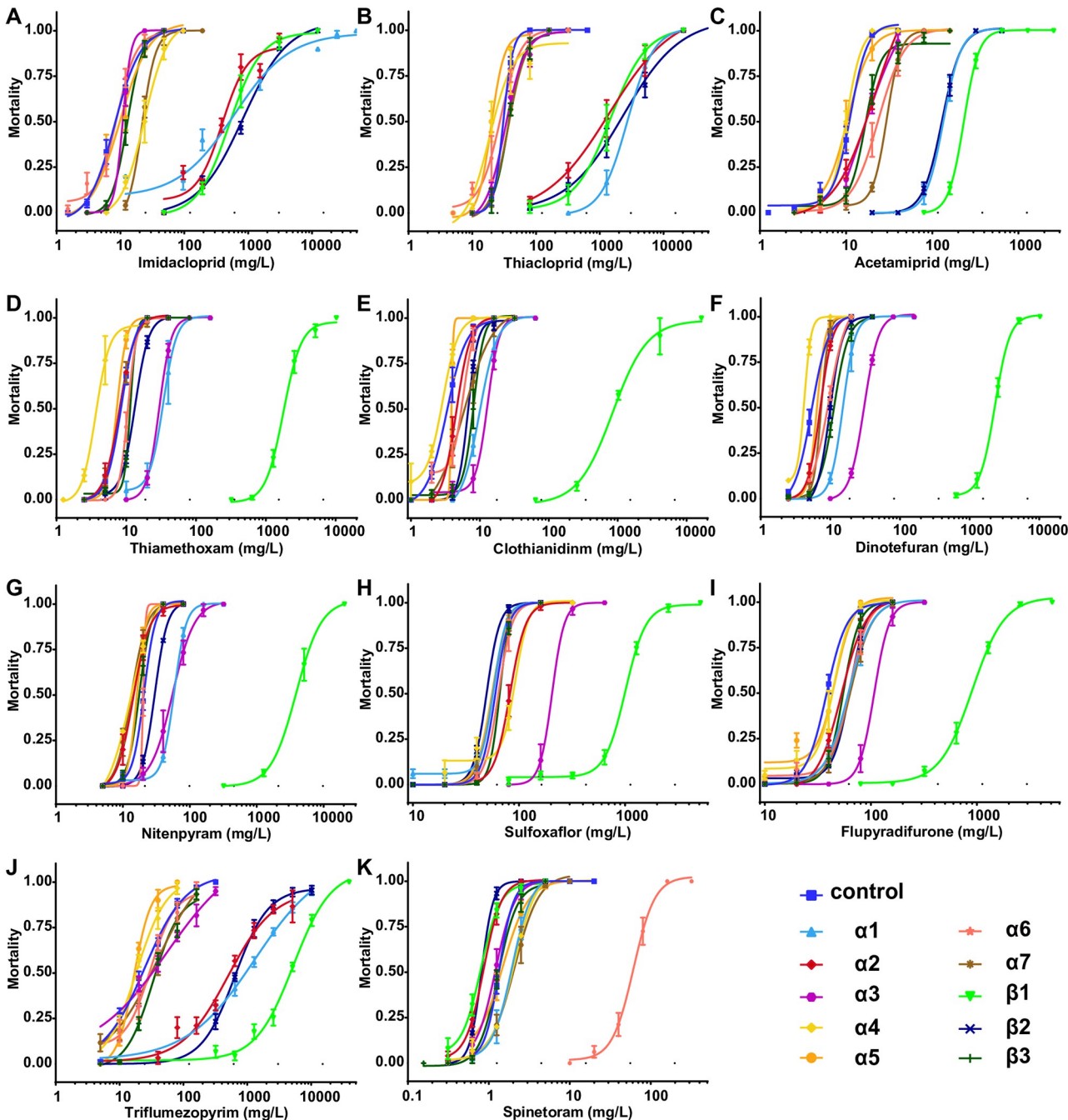

**Fig 2. Nonlinear log-dose mortality data for tested insecticides against ten *Drosophila* nAChR homozygous mutants, including eight null alleles and two point mutation alleles ($\alpha 4^{T227M}$ and $\beta 1^{R81T}$).** Mortality (0–1 means 0–100% in terms of percentage) of control and mutant female adults after 48 hours of exposure to increasing concentrations of insecticides. Error bars represent standard deviations.

high resistance to these insecticides. Therefore, thermogenetic activation of some *nAChR*-expressing neurons in a short time window phenocopies the action of insecticides in target pests, which demonstrates that in vivo pharmacological activation of these subunit-containing nAChRs leads to toxicity and ultimately death.

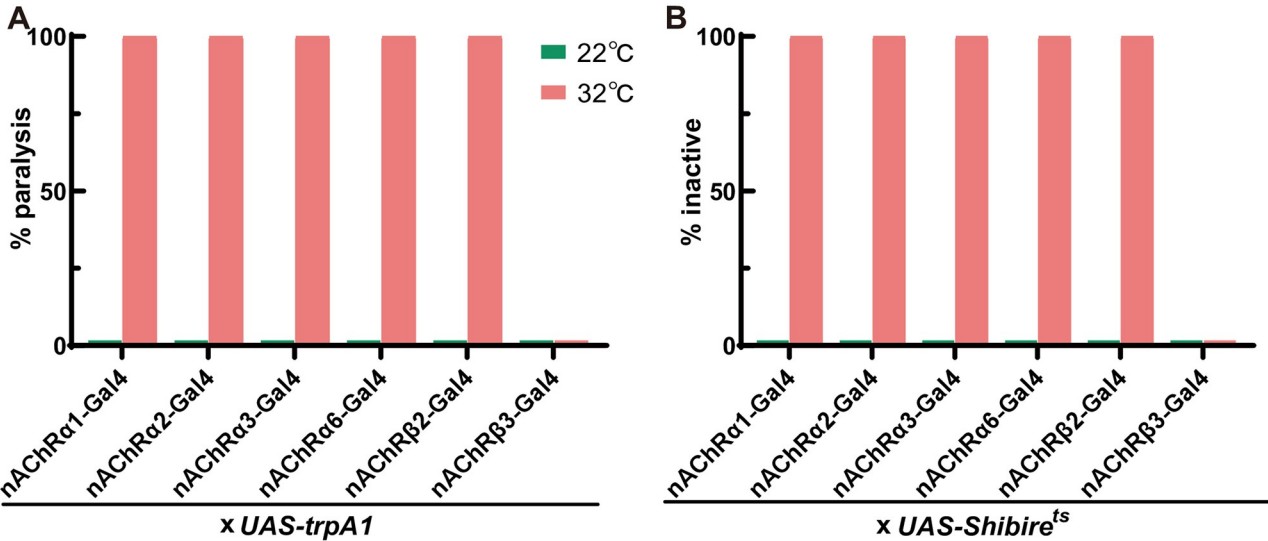

**Fig 3. Effects of artificial neuronal activation and inhibition in various *nAChR*-expressing neurons.** (A) Thermogenetic activation of five *nAChR*-expressing neurons using *UAS-trpA1*-induced paralysis behavior. (B) Thermogenetic silencing of five *nAChR*-expressing neurons using *UAS-Shibire*ts decreased activity. n = 30–50.

The poisoning symptoms associated with triflumezopyrim are distinct from other insecticides that act on nAChRs since it inhibits rather than activates insect nAChRs. There are no neuroexcitatory symptoms after treatment with triflumezopyrim; in contrast, triflumezopyrim induces lethargic poisoning characterized by slow but coordinated leg movements, and insects become less responsive to stimuli over time [20]. Thus, we chose to use *UAS-Shibire*ts to inhibit *nAChR-expressing* neurons [21]. As expected, *nAChRα1*2A-GAL4, *nAChRα2*2A-GAL4 and *nAChRβ2*2A-GAL4 neurons produced "sluggish" behavior rather than hyperactivity (Fig 3B). The flies exhibited almost no translational or rotational body movement (S1 Video). Silencing of *nAChRα3*2A-GAL4 and *nAChRα6*2A-GAL4 neurons also produced similar behaviors, further confirming that the α3- and α6-containing nAChRs cannot be blocked by triflumezopyrim; otherwise, both mutants would show resistance in bioassays.

## Expression patterns of nAChRs in KO mutants

We confirmed that the KO coding regions were not detected or barely detectable with real-time PCR quantification (Fig C in S1 Text). There was no large difference in the expression levels of each subunit in these mutant flies except for β3, which was relatively less transcribed than the other genes. For the α1 heterozygous mutant, the mRNA levels of all subunits were almost the same as those of the wild-type control.

## Discussion

The Insecticide Resistance Action Committee (IRAC) classifies neonicotinoids, sulfoximines, butenolides and mesoionics according to their chemical similarity relations into subgroups 4A, 4C, 4D and 4E, respectively. However, our results showed that sulfoxaflor and flupyradifurone may mainly act on the same nAChR subtype, which consists of α3 and β1 subunits (Fig 4), although other subunits may also be involved considering genetic redundancy. More importantly, we found that neonicotinoids act on distinct nAChR subtypes and that such selectivity is not dependent on the aromatic heterocyclic (A) or the electron-withdrawing nitro or cyano moiety (X-Y), which is considered the key toxophore. Interestingly, the ring systems

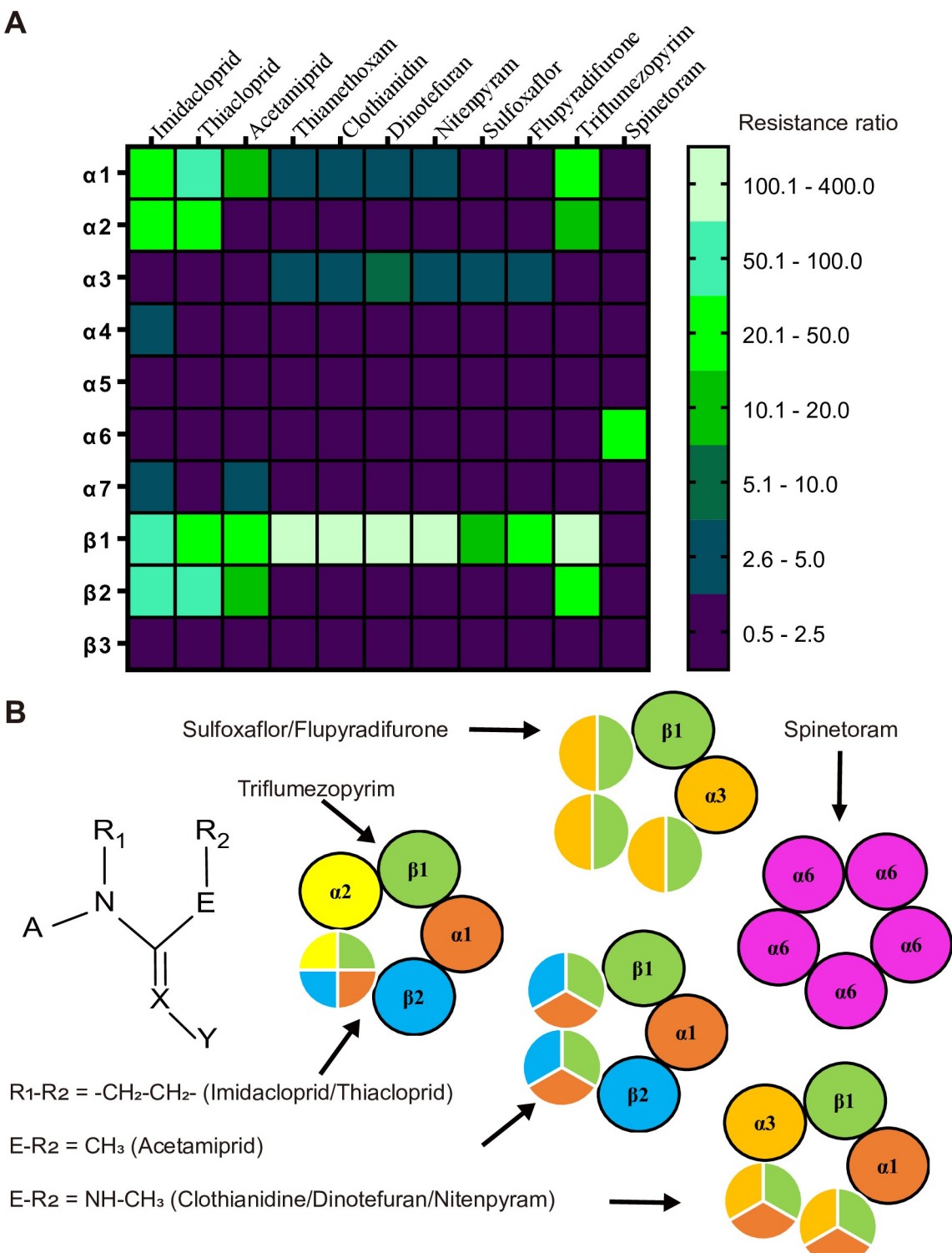

**Fig 4. Resistance patterns of the tested insecticides on different nAChR mutants (A) and proposed target receptor subtypes for neonicotinoids and others (B).** Various resistance ratios are grouped and represented as different colors in the heatmap. Thiamethoxam is considered a prodrug of clothianidin and thus is not listed in the structural formula.

and the $R_2$ substituents in the open-chain structures are the determining factors (Fig 4). For example, the α1, α2, β1 and β2 mutants showed similar levels of resistance to imidacloprid and thiacloprid (both have a five-membered ring), indicating that they mainly act on the same α1/α2/β1/β2 pentamer (Fig 4B). This finding is consistent with previous ex vivo recording results [22] and two recent reconstituted studies, which showed that both drugs act as partial agonists on the α1/α2/β1/β2 nAChR [7, 8]. Acetamiprid is structurally similar to thiacloprid with the cyanoimine pharmacophore, although the acyclic configuration changes its molecular target in vivo. It may act on the α1/β1/β2 nAChR, and again, electrophysiological studies have already indicated that acetamiprid is nearly a full agonist [22]. Moreover, its potency on the recombinant louse α1/α2/β1/β2 nAChR is approximately 860-fold lower than that of thiacloprid [8]. Although thiamethoxam has a six-membered ring, it is a prodrug without intrinsic nAChR activity until metabolized to the active form clothianidine in plants and insects [23]. Therefore, thiamethoxam, clothianidine, dinotefuran and nitenpyram can be considered the same type, which has N-methyl substitutions in the $R_2$ position and mainly acts on the α1/α3/β1 nAChR (Fig 4B). Neonicotinoids are traditionally divided into nitroimines ($NNO_2$), nitromethylenes (CHNO2) or cyanoimines (NCN), although our findings indicate that they should be classified according to their major nAChR subtype targets.

Despite the widespread use of neonicotinoids for almost three decades, the first and only field-evolved target-site resistance mutation (R81T in nAChRβ1) was reported in 2011, and it has only been found in two species to date [13, 14]. This unusual phenomenon is consistent with our findings that the seven neonicotinoids actually act on multiple receptor types in vivo and that only the β1 mutant caused high resistance to all neonicotinoids. New nicotine-mimicking insecticides, such as sulfoxaflor and flupyradifurone, mainly act on another nAChR subtype that is distinct from neonicotinoids (Fig 4), indicating their potential use in insecticide resistance management.

Electrophysiological studies with native tissues or recombinant receptors showed that low concentrations of neonicotinoids can block nAChR, while higher concentrations can activate the receptor [7,24]. Therefore, it is still unclear whether insecticidal activity is the consequence of nAChR inhibition or activation in vivo. We found that transient artificial activation rather than inhibition of *nAChR*-expressing neurons was sufficient to induce neonicotinoid-like poisoning symptoms in flies (Fig 3). Thus, the overall effect of neonicotinoids is neuronal depolarization by activation of nAChR, which is more physiologically relevant.

Triflumezopyrim is the first member of a new class of mesoionic insecticides that act via inhibition of the orthosteric binding site of the nAChR [20]. We found that the α1/α2/β1/β2 nAChR could be its major target, similar to imidacloprid and thiacloprid, and all these mutants showed high resistance to triflumezopyrim (Fig 4A). This finding is consistent with radioligand binding results showing that triflumezopyrim potently displaced [³H]imidacloprid with a Ki value of 43 nM based on membrane preparations from the aphid [20]. Thermogenetic inhibition of neurons expressing α1, α2 and β2 also mimicked lethargic intoxication symptoms (Fig 3B). Thus, to maintain the durability and effectiveness of this new powerful tool for the control of hopper species in rice, it is critical to avoid repeated use of triflumezopyrim with imidacloprid and thiacloprid.

Spinosyns, including spinosad and spinetoram, have been shown to act on a population of nAChRs that are not targeted by neonicotinoids, and the binding site is also distinct from the orthosteric site [4]. The α6 subunit has been proposed as the main target of spinosyns since the field-evolved resistance to spinosad is associated with loss-of-function mutations of α6 loci in many pest insects [25–32]. However, the involvement of other subunits, such as α5 and α7, which are phylogenetically close to α6, has not been clarified (Fig D in S1 Text). Previous reports showed that α5 and α7 can form functional homomeric and heteromeric channels in

vitro while α6 can only form heteromeric channels with α5 or α5/α7 together [33, 34]. We then wondered whether there was genetic redundancy among these evolutionarily conserved subunits. We found that the α5, α7 and α5/α7 double mutants were all sensitive to spinetoram (Table K in S1 Text), indicating that spinosyns may exclusively act on the α6 homomeric nAChR, which is consistent with a recent report using spinosad [35]. Thermogenetic activation of *α6*-expressing neurons also induced spinosyn-like poisoning symptoms in flies.

Our current knowledge about the subunit composition of insect nAChRs is very limited. Immunoprecipitation data with subunit-specific antibodies showed that *Drosophila* α3 and β1 coassemble within the same receptor complex [36]. Further studies from the same group indicated that α1/α2/β2 and β1/β2 may coassemble into the same receptor complex [37]. Similar studies using the brown planthopper suggested that there are two populations of nAChRs that contain *Drosophila*-equivalent subunit combinations α1/α2/β1 and α3/β1/β2 [38]. These previous findings are partially confirmed by the present results because α3/β1, α1/α3/β1, α1/β1/β2 and α1/α2/β1/β2 could be the major receptor subtypes for the tested insecticides, indicating that the β1 subunit could be an indispensable component for all heteromecic pentamers (Fig 4). In addition, we noticed that for some insecticides, different subunit mutations contribute in an asymmetrical manner to resistance (Fig 4A). Therefore, there could be functional redundancy between some *α*-type subunits, and we cannot exclude the existence of other potential receptor subtypes, such as α1/β1 and α3/β1/β2. The diversity of insect nAChRs and their druggability make them an extremely important target for insecticide development.

Growing evidence indicates that sublethal doses of neonicotinoids, such as imidacloprid, thiamethoxam and clothianidin negatively affect wild and managed bees, which are important pollinators in ecosystems and agriculture [39–41]. They reduce reproduction and colony development, perhaps by impairing the foraging, homing and nursing behaviors of bees [42]. These severe sublethal effects have led to heavy restrictions on the use of the above three neonicotinoids in Europe to protect bee pollinators [43]. Sulfoxaflor and flupyradifurone are potential alternatives for neonicotinoids; however, their risk to bees is controversial [44–46]. Therefore, it is critical to understand the mode of action of these insecticides inside bees. The core groups of nAChR subunits are highly conserved among different insects spanning ~300 million years of evolution [47], which is likely due to their essential roles in the nervous system. Most *Drosophila* nAChR subunit genes (except *α5* and β3) have one-to-one orthologs in other insects, including honeybees and bumblebees (Fig D in S1 Text), and the sequence identities between orthologs are also high (Table L in S1 Text). Thus, the expression, assembly and function of these receptors could be conserved between flies and bees, suggesting that our results will enable further studies about the ecotoxicology and risk assessment of these nAChR modulators.

## Materials and methods

### Insecticides

The following were purchased commercially: imidacloprid (600 g/LSC, Bayer CropScience, Germany), thiamethoxam (70%GZ, Syngenta, China), clothianidin (48%SC, HeNan Hansi crop protection, China), dinotefuran (20%SG, Mitsui Chemicals, Japan), nitenpyram (30% WG, ZinGrow, China), acetamiprid (20%SP, Noposion, China), thiacloprid (40%SC, Limin Chemical, China), sulfoxaflor (22%SC, Dow AgroSciences, USA), flupyradifurone (17%SC, Bayer CropScience, Germany), triflumezopyrim (10%SC, DuPont, USA), spinetoram (60 g/ LSC, Dow AgroSciences, USA) and Triton X-100 (Sangon Biotech, China).

## Fly strains

Flies were maintained and reared on conventional cornmeal agar molasses medium at $25 \pm 1°C$ and $60\% \pm 10\%$ humidity with a photoperiod of 12 hours light:12 hours night. For experiments using *UAS-trpA1* and *UAS-Shibire^ts* transgenes, flies were reared at 21°C. The following strains were sourced from the Bloomington Stock Center (Indiana University): vas-cas (#51323), *UAS-trpA1* (#26263), and *UAS-Shibire^ts* (44222). All nAChR KO mutants and KI-Gal4 strains were gifts from Dr. Yi Rao (Deng et al., 2019) (Peking University). The *w^1118* strain used for outcrossing was used as a wild type for the insecticide bioassays.

We generated the nAChRβ1^R81T mutant by CRISPR/Cas9 genome editing. The gRNA sequence (3 L:4433329~4433352, ATCAAACGTTTGGTTAACTTTAG) was designed with flyCRISPR Target Finder (https://flycrispr.org/target-finder/) and cloned into the pDCC6 plasmid (Addgene #59985). A 110 bp ssODN (single-strand oligodeoxynucleotide) was custom-synthesized as the donor template to replace the targeted genomic region. This ssODN contained three nucleotide changes, with two (CG to AC) conferring the R81T mutation and one synonymous mutation (G to A) to prevent recleavage from Cas9 after incorporation. Both the gRNA plasmid and ssODN were microinjected into the embryos of *vas-cas* flies (BL #51323). The crossing and selection scheme is shown in Fig A in S1 Text.

## Insecticide bioassays

Three- to five-day-old and uniformly sized adult females were used in insecticide bioassays to assess the susceptibility of different fly strains. The testing method was modified from the IRAC susceptibility test method 026 (https://irac-online.org/methods/). Briefly, the required serial dilutions of insecticide solution were prepared in 200 g/L sucrose using formulated insecticides. Approximately 5 ml of insecticide solution is required for each concentration. A piece of dental wick (1 cm) was placed in a standard *Drosophila* vial and treated with 800 μL of 20% aqueous sucrose with or without insecticide. Ten flies of each genotype were transferred into vials, with 3–6 vials for each concentration, and each genotype was repeated at least 3 times for every tested insecticide. The vials were kept upside down until all flies became active to avoid flies becoming trapped in the dental wick. The bioassay was assessed after 48 h, and dead flies and seriously affected flies displaying no coordinated movement that were unable to walk up the vial or get to their feet were cumulatively scored as 'affected'. The $LC_{50}$ values were calculated by probit analysis using Polo Plus software (LeOra Software, Berkeley, CA, USA). Nonlinear log dose–response curves were generated in GraphPad Prism 8.21 (GraphPad Software Inc., La Jolla, CA, USA).

## Thermogenetic activation and silencing assays

Flies for TRPA1-mediated thermogenetic activation and Shibire-mediated silencing experiments were collected upon eclosion and reared in vials containing standard food medium at 21°C for 5–8 days. For thermogenetic activation with the UAS-trpA1 transgene, 10 flies were transferred to new empty vials by gentle inspiration, and then the assays were performed at 23°C and 32°C for 10 minutes. Each genotype was repeated for at least 5 times. The percentage of paralysis behavior in which the animal lies on its back with little effective movement of the legs and wings was measured. For the silencing assays, the UAS-Shibire^ts transgene was used and flies were also transferred to fly vials at 23°C and 32°C for 10 minutes.

## Real-time quantitative PCR

The relative transcription levels of *nAChRs* in different KO mutants were examined using real-time quantitative PCR performed with a CFX96TM Real-Time PCR System (Bio–Rad,

Hercules, USA). Total RNA was isolated with TRIzol reagent according to the manufacturer's instructions. Residual genomic DNA was removed by RQ1 RNase-Free DNase (Promega, Madison, USA). Total RNA was reverse transcribed to cDNA with EasyScript First-Strand cDNA Synthesis SuperMix (Transgene, Beijing, China). qPCR with gene-specific primers was performed with ChamQ Universal SYBR qPCR Master Mix (Vazyme, Nanjing, China) to investigate the relative expression levels of different *nAChRs*. *RpL32* (ribosomal protein L32) was used as an internal control. The relative expression of *nAChRs* was normalized to the reference (RpL32) using the $2^{-\Delta\Delta CT}$ method. The primers used are listed in Table M in S1 Text.

## Fecundity and development assays

Ten pairs of freshly emerged couples of wild-type control and β1$^{R81T}$ mutant were transferred into vials containing normal food for 72 hours. These files were then transferred into a new dish that was used for the egg-laying assay. The numbers of eggs laid in each dish were recorded after 24 hours. To calculate the larval to pupal rate, 60 second-instar larvae were collected and transferred into a new vial as one group. The numbers of pupae in each vial were recorded after 7 days in an incubator. Each genotype was repeated at least three times in duplicate.

## Climbing assay

Approximately three-day-old male flies were collected with $CO_2$ anesthesia into groups of 10 and then allowed to recover for 2 days. A climbing tube consisted of two vials with 90 mm height and 20 mm diameter. The flies were filmed for 30 seconds with a SONY HDR-CX900E camera. The climbing index (percentage of flies in the upper half of the vial) was determined at 5 second intervals after the flies had been tapped down to the bottom of the vials.

## Phylogenetic analysis

The following representative species from a variety of orders were selected: *Apis mellifera* (honey bee), *Tribolium castaneum* (red flour beetle), *Myzus persicae* (green peach aphid), *Bombyx mori* (silk worm), *Bombus terrestris* (bumble bee) and *Drosophila melanogaster* (fruit fly). To identify the orthologs of the *D. melanogaster* nAChR subunits, we searched against NCBI non-redundant protein database using BLASTP. We renamed *M. persicae* and *B. terrestris* nAChR subunit proteins according to their closest orthologs. All the amino acid sequences were aligned by Clustal X. A neighbor-joining tree was performed by MEGA 11 with default parameters, 1000 bootstrap replications, and substitution with JTT model and visualized by Evolview (https://www.evolgenius.info//evolview/).

## Supporting information

**S1 Video. Effects of thermogenetic activation and inhibition in *nAChRα1*- expressing neurons.** The following transgenes were used: *nAChRα1*$^{2A-GAL4}$ > *UAS-trpA1*; *nAChRα1*$^{2A-GAL4}$ > *UAS-Shibire*$^{ts}$. Other *nAChR* KI-Gal4 strains like n*AChRα2*$^{2A-GAL4}$, *nAChRα3*$^{2A-GAL4}$, *nAChRα6*$^{2A-GAL4}$ and *nAChRβ2*$^{2A-GAL4}$ also produced similar behaviors when stimulated under 32˚C.
(MP4)

**S1 Text.** Fig A in S1 Text. The crossing schemes to establish the *nAChRβ1*$^{R81T}$ knock-in line. The HDR event was isolated by imidacloprid selection and confirmed by PCR. The *vas-Cas9* (3XP3 RFP) was removed by the absence of red fluorescence in eyes. Fig B in S1 Text. Effects of *nAChRβ1*$^{R81T}$ point mutation on number of eggs laid (A), pupation rate of larvae (B) and

negative geotaxis behavior (C). Fig C in S1 Text. Expression patterns of the nAChR genes in different KO mutants. Fig D in S1 Text. Phylogenetic relationships of core groups of nAChR subunits from 6 representative insect species including *Apis mellifera* (honey bee), *Tribolium castaneum* (red flour beetle), *Myzus persicae* (green peach aphid), *Bombyx mori* (silk worm), *Bombus terrestris* (bumble bee) and *Drosophila melanogaster* (fruit fly). The colorful dots at the nodes of the branches represent the values of bootstrap support for each branch. The *D. melanogaster* FMRFamide receptor (DmFR) was used as an outgroup. The sequence accession numbers are shown in Table N in S1 Text. Table A in S1 Text. Log dose probit mortality data and resistance ratios for imidacloprid. Table B in S1 Text. Log dose probit mortality data and resistance ratios for thiacloprid. Table C in S1 Text. Log dose probit mortality data and resistance ratios for acetamiprid. Table D in S1 Text. Log dose probit mortality data and resistance ratios for thiamethoxam. Table E in S1 Text. Log dose probit mortality data and resistance ratios for clothianidin. Table F in S1 Text. dose probit mortality data and resistance ratios for dinotefuran. Table G in S1 Text. Log dose probit mortality data and resistance ratios for nitenpyram. Table H in S1 Text. Log dose probit mortality data and resistance ratios for flupyradifurone. Table I in S1 Text. Log dose probit mortality data and resistance ratios for sulfoxaflor. Table J in S1 Text. Log dose probit mortality data and resistance ratios for triflumezopyrim. Table K in S1 Text. Log dose probit mortality data and resistance ratios for spinetoram. Table L in S1 Text. Sequence identities between *Drosophila* nAChR subunits and corresponding orthologs in other insects. Table M in S1 Text. Primers used in qPCR analysis. Table N in S1 Text. The accession numbers of sequences used in Fig D in S1 Text.
(DOCX)

## Acknowledgments

We thank Yi Rao (Peking University) and the Bloomington Drosophila Stock Center for fly stocks.

## Author Contributions

**Conceptualization:** Jia Huang.

**Data curation:** Wanjun Lu, Zhihan Liu, Xinyu Fan, Xinzhong Zhang.

**Formal analysis:** Xinzhong Zhang, Xiaomu Qiao, Jia Huang.

**Funding acquisition:** Xiaomu Qiao, Jia Huang.

**Investigation:** Xiaomu Qiao, Jia Huang.

**Methodology:** Xinzhong Zhang, Xiaomu Qiao.

**Project administration:** Jia Huang.

**Supervision:** Jia Huang.

**Writing – original draft:** Xiaomu Qiao, Jia Huang.

**Writing – review & editing:** Jia Huang.

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
