## [Decision Letter · Decision Letter 0]

29 Nov 2021

Dear Dr Huang,

Thank you very much for submitting your Research Article entitled 'Nicotinic modulation insecticides act on diverse receptor subtypes with distinct subunit compositions' to PLOS Genetics.

The manuscript was fully evaluated at the editorial level and by independent peer reviewers. The reviewers appreciated the attention to an important problem, but raised some substantial concerns about the current manuscript. Based on the reviews, we will not be able to accept this version of the manuscript, but we would be willing to review a much-revised version. We cannot, of course, promise publication at that time.

If you decide to revise the manuscript for further consideration at PLOS Genetics, please aim to resubmit within the next 60 days, unless it will take extra time to address the concerns of the reviewers, in which case we would appreciate an expected resubmission date by email to plosgenetics@plos.org.

[LINK]

We are sorry that we cannot be more positive about your manuscript at this stage. Please do not hesitate to contact us if you have any concerns or questions.

Yours sincerely,

Subba Reddy Palli, Ph.D.

Associate Editor

PLOS Genetics

Gregory P. Copenhaver

Editor-in-Chief

PLOS Genetics

Reviewer's Responses to Questions

**Comments to the Authors:**

Reviewer #1: This manuscript describes the use of 10 mutant lines of Drosophila to investigate the role of different nAChR subunit subtypes in mediating the response to insecticidal agonists of the nAChR. While the work performed in the study is conceptually very simple, the results obtained are interesting and provide new information on this topic. However, many overblown and misleading claims are made in the manuscript that are not supported by the data, these significantly diminish the quality of the presented paper. The manuscript also suffers from several issues in regards its presentation and description of the work done.

Major comments

The quality of the written English in the manuscript is poor in places and requires revision prior to publication.

In the title, abstract and elsewhere, the authors use the term: ‘nicotinic modulation insecticides’ this is at best ambiguous and unclear. Please avoid use of this term and replace with a suitable alternative, such as ‘insecticides acting on the nicotinic acetylcholine receptor’.

Line 47/48. The authors state: ‘Although all act via insect nicotinic acetylcholine receptors, the mode of action is unclear.’ This is misleading, actually a great deal of work has been done on the MOA of these insecticides and their action at the nAChR. Thus this claim should be removed or modified.

Line 80: The authors state: ‘Thus, neonicotinoids account for 24% of the global insecticide’, however at line 44 they state: ‘The neonicotinoids and spinosyns make up about 27% of the insecticides by world market value’. This discrepancy should be corrected.

Line 127/128: The authors state that knock out of two of 10 nAChR subunits (α4 and β1) was homozygous lethal, however they then start talking about the introduction of two mutations into these subunits. The link here is unclear. Furthermore why not use one of the many genetic tools available in Drosophila (i.e. balancers etc.) to investigate these subunits in the heterozygous form? If this is actually what they did please make this significantly more obvious (and how this was achieved in the results).

Lines 219: The authors state: ‘However, our results clearly showed that sulfoxaflor and flupyradifurone may specifically act on the same nAChR subtype which consists of α3 and β1 subunits’. I don’t see how the authors can make this claim based on the work performed. Evidence is presented to suggest that nAChRs containing these subunits are sensitive to these insecticides, but conclusions on which other nAChR subunit types co-assemble with them cannot be established based on the work done. What the data does, and does, not show needs to be much more clearly and carefully defined.

Lines 248-252: The authors state: ‘Such unusual phenomenon can be partially explained by our findings that the seven neonicotinoids have at least three distinct molecular targets in vivo. To some extent, the continuous use of different neonicotinoids is a kind of spontaneous insecticides rotations, which has been proven to be effective in mitigating or delaying resistance’. This is a very bold claim that I believe to be misleading, inaccurate and not supported by phenotypic investigation of the impact of the known resistance mutations. Specifically, R81T has been shown to confer resistance to multiple neonicotinoid insecticides, and this finding directly contradicts this conclusion.

Line 277: The authors claim: ‘The α6 subunit has been proposed as the main target of spinosyns since the resistance to spinosad in many insects is associated with loss-of-function mutations in the α6 gene [24], however, whether other subunits are involved is still unknown.’ Again this is a somewhat misleading statement that discredits work done by numerous labs on this topic. There is unequivocal evidence that the α6 subunit IS the main target of spinosyns and absolutely no evidence at all that any other subunits are involved. Thus the work presented on this topic by the current authors is not novel and any claims that it is are inaccurate and should be moderated.

Lines 312: The authors state: ‘Since most Drosophila nAChR subunit genes (except α5 and β3) have one-to-one orthologs in the honeybee and bumblebee genomes [7], the expression and assembly of receptors could be conserved between flies and bees, suggesting that our results will enable further studies about the ecotoxicology and risk assessment for these nAChR modulators.’ This is a very big jump, just because two different insect species have orthologous genes does not mean that inferences on one can be applied to another, especially when the two species are separated by millions of years of evolution and have completely different life histories!

Reviewer #2: Lu et al. investigated the impact and toxicity of modulators (IRAC MoA group 4) of the nicotinic acetylcholine receptor (nAChR) in transgenic flies either lacking individual receptor subtypes, or – in two cases – flies with mutated subunits, because their knock-out was lethal. In total the authors investigated 11 insecticides including neonicotinoids, butenolides, sulfoximines, mesoionics and spinosyns. They convincingly demonstrated by RT-qPCR that the different mutant fly lines lack the respective subunits they are supposed to lack. The authors conducted bioassays with each transgenic line in comparison to wildtype flies and calculated resistance ratios for each insecticide. Furthermore, the authors conducted climbing assays with each fly line and mimicked the symptomology of poisoning for some insecticides by thermogenetic activation of nAChR expressing neurons. Finally, they conducted fecundity and development assays with flies expressing a mutant R81T ß1 subunit, which is known to confer neonicotinoid resistance in aphids.

The study is well executed and the methods are described in enough detail. The introduction and discussion are well referenced and there is not much to criticize. Many of the presented findings are novel and advance the knowledge in the field of insect nAChR modulator mode of action. However, employing Drosophila as a proxy may not always reflect the same effects one would expect in pest insects or beneficials targeted by these insecticides, especially since dipterans and hymenopterans diverged approx. 300 mya. However, as the authors outlined, in many sequenced insect genomes 1:1 orthologs for the different nAChR subunits are present, thus justifying some of the claims made by the authors. It is of particular interest that the ß1 subunit seems most crucial for all insecticides tested, except spinetoram. I have only a few minor points I want the authors to address and listed below.

1) I would suggest to slightly change the title as “nicotinic modulation insecticides” sounds a bit awkward. I think “Nicotinic acetylcholine receptor modulator insecticides act on…” would be better.

2) L25: add “bee” pollinators

3) L27: I do not think that the MoA of these insecticides is unclear for 30 years. Please reword, because many good papers were published on the molecular MoA of these insecticides.

4) L69: I do not agree that for MANY insecticides the exact molecular targets remain elusive. Please tone down.

5) L245: Imidacloprid was the first neonicotinoid launched to the market in 1991, so these insecticides are around for three decades, not four.

6) L351: Insecticide bioassay – how many replicated were tested?

7) L402: Climbing assay – how many replicated were tested?

**Have all data underlying the figures and results presented in the manuscript been provided?**

Reviewer #1: Yes

Reviewer #2: Yes

PLOS authors have the option to publish the peer review history of their article (what does this mean?). If published, this will include your full peer review and any attached files.

Reviewer #1: No

Reviewer #2: No

---

## [Decision Letter · Decision Letter 1]

23 Dec 2021

Dear Dr Huang,

We are pleased to inform you that your manuscript entitled "Nicotinic acetylcholine receptor modulator insecticides act on diverse receptor subtypes with distinct subunit compositions" has been editorially accepted for publication in PLOS Genetics. Congratulations!

Yours sincerely,

Subba Reddy Palli, Ph.D.

Associate Editor

PLOS Genetics

Gregory P. Copenhaver

Editor-in-Chief

PLOS Genetics

Comments from the reviewers (if applicable):

Reviewer's Responses to Questions

**Comments to the Authors:**

Reviewer #1: The authors have adressed or rebutted all my comments and the revised manuscript is signifcantly improved. I have no further comments.

Reviewer #2: All points raised in my review were adequately addressed by the authors.

**Have all data underlying the figures and results presented in the manuscript been provided?**

Reviewer #1: None

Reviewer #2: Yes

PLOS authors have the option to publish the peer review history of their article (what does this mean?). If published, this will include your full peer review and any attached files.

Reviewer #1: No

Reviewer #2: No

**Data Deposition**

http://datadryad.org/submit?journalID=pgenetics&manu=PGENETICS-D-21-01445R1

**Press Queries**

---

## [Editor Report · Acceptance letter]

14 Jan 2022

PGENETICS-D-21-01445R1 

Nicotinic acetylcholine receptor modulator insecticides act on diverse receptor subtypes with distinct subunit compositions 

Dear Dr Huang, 

We are pleased to inform you that your manuscript entitled "Nicotinic acetylcholine receptor modulator insecticides act on diverse receptor subtypes with distinct subunit compositions" has been formally accepted for publication in PLOS Genetics! Your manuscript is now with our production department and you will be notified of the publication date in due course.

With kind regards,

Zsanett Szabo

PLOS Genetics

On behalf of:
